# 4-Coumaroyl-CoA ligases in the biosynthesis of the anti-diabetic metabolite montbretin A

Frederick G. Sunstrum[1], Hannah L. Liu[1], Sharon Jancsik[1], Lufiani L. Madilao[1], Joerg Bohlmann[1,2,3]*, Sandra Irmisch[1]*

**1** Michael Smith Laboratories, University of British Columbia, Vancouver, British Columbia, Canada, **2** Department of Botany, University of British Columbia, Vancouver, British Columbia, Canada, **3** Department of Forest and Conservation Sciences, University of British Columbia, Vancouver, British Columbia, Canada

\* bohlmann@msl.ubc.ca (JB); sirmisch@msl.ubc.ca (SI)

## Abstract

### Background

Montbretins are rare specialized metabolites found in montbretia (*Crocosmia x crocosmii-flora*) corms. Montbretin A (MbA) is of particular interest as a novel therapeutic for type-2 diabetes and obesity. There is no scalable production system for this complex acylated flavonol glycoside. MbA biosynthesis has been reconstructed in *Nicotiana benthamiana* using montbretia genes for the assembly of MbA from its various different building blocks. However, in addition to smaller amounts of MbA, the therapeutically inactive montbretin B (MbB) was the major product of this metabolic engineering effort. MbA and MbB differ in a single hydroxyl group of their acyl side chains, which are derived from caffeoyl-CoA and coumaroyl-CoA, respectively. Biosynthesis of both MbA and MbB also require coumaroyl-CoA for the formation of the myricetin core. Caffeoyl-CoA and coumaroyl-CoA are formed in the central phenylpropanoid pathway by acyl activating enzymes (AAEs) known as 4-coumaroyl-CoA ligases (4CLs). Here we investigated a small family of montbretia AAEs and 4CLs, and their possible contribution to montbretin biosynthesis.

### Results

Transcriptome analysis for gene expression patterns related to montbretin biosynthesis identified eight different montbretia AAEs belonging to four different clades. Enzyme characterization identified 4CL activity for two clade IV members, Cc4CL1 and Cc4CL2, converting different hydroxycinnamic acids into the corresponding CoA thioesters. Both enzymes preferred coumaric acid over caffeic acid as a substrate *in vitro*. While expression of montbretia AAEs did not enhance MbA biosynthesis in *N. benthamiana*, we demonstrated that both Cc4CLs can be used to activate coumaric and caffeic acid towards flavanone biosynthesis in yeast (*Saccharomyces cerevisiae*).

### Conclusions

Montbretia expresses two functional 4CLs, but neither of them is specific for the formation of caffeoyl-CoA. Based on differential expression analysis and phylogeny Cc4CL1 is most

**Data Availability Statement:** All relevant data are within the paper and its Supporting information files.

**Funding:** The research was supported with funds from the GlycoNet Networks of Centres of Excellence (to J.B.), the Natural Sciences and Engineering Research Council of Canada (NSERC Discovery Grant to J.B.), NSERC CGS-M fellowship (to F.G.S.), and the Alexander von Humboldt Foundation through a Feodor Lynen Research Fellowship (to S.I.). The funders had no role in study design, data collection and analysis, decision to publish, or preparation of the manuscript.

**Competing interests:** The authors have declared that no competing interests exist.

likely involved in MbA biosynthesis, while Cc4CL2 may contribute to lignin biosynthesis. Both Cc4CLs can be used for flavanone production to support metabolic engineering of MbA in yeast.

## Introduction

The large chemical space of specialized (i.e., secondary) metabolites produced by plants contains some of the most powerful medicinally active compounds. Montbretia (*Crocosmia x crocosmiiflora*), a member of the Iris family (Iridaceae), produces the flavonoid molecule montbretin A (MbA) and a series of related molecules MbB and MbC [1, 2] (Fig 1). MbA is a complex acylated flavonol glycoside, defined as myricetin 3-*O*-(glucosyl-6'-*O*-caffeoyl)-1,2-β-D-glucosyl 1,2-α-L-rhamnoside 4'-*O*-α-L-rhamnosyl 1,4-β-D-xyloside. MbA inhibits human pancreatic α-amylase (HPA) in a highly efficient and selective manner, thereby slowing starch digestion in the intestine and reducing post-prandial blood glucose levels [3]. This unique activity and specificity make MbA a candidate for its development into a novel pharmaceutical or nutraceutical to treat or prevent type-2 diabetes and obesity [2, 4]. MbA has successfully passed animal trials with diabetic rats and is now entering human clinical trials [3]. A sustainable and scalable production system is needed to develop MbA as a widely available pharmaceutical or nutraceutical. Currently, the only known source for MbA are the underground storage and reproductive organs, called corms, of montbretia [5]. The rare taxonomic occurrence and low amounts of MbA in montbretia corms [2, 5] together with the lack of large-scale cultivation, currently render montbretia inadequate for farm-based MbA production. Large-scale chemical synthesis of MbA is also not feasible due to its complex chemical structure. MbA production by means of synthetic biology or metabolic engineering of a recombinant plant or microbial system appears to be a viable approach [5–8].

To enable metabolic engineering of MbA production, we are investigating the genes and enzymes of MbA biosynthesis in montbretia. We recently elucidated the assembly of the complete MbA molecule from its different building blocks, specifically myricetin, UDP-glucose, UDP-rhamnose, UDP-xylose, and caffeoyl-CoA. Five UDP-dependent glycosyltransferases (UGTs) and a BAHD-acyltransferase (AT) have been identified to catalyze MbA assembly in a linear six-step biosynthetic pathway [5, 6, 8]. The third step in MbA biosynthesis is the acylation of myricetin glycosyl rhamnoside (MRG) into mini-MbA catalyzed by CcAT. This enzyme is not specific for its acyl donor and accepts different hydroxycinnamoyl-CoA thioesters as substrates. The incorporation of caffeoyl-CoA, coumaroyl-CoA, or feruloyl-CoA leads to the formation of MbA, montbretin B (MbB), or montbretin C (MbC), respectively (Fig 1). The *meta*-hydroxy group of the caffeoyl moiety in MbA is essential for the biological activity of HPA inhibition; MbB and MbC are not efficient inhibitors of HPA due to the absence and methylation, respectively, of this *meta*-hydroxy group [2, 4].

We are exploring both yeast (*Saccharomyces cerevisiae*) and *Nicotiana benthamiana* as hosts for metabolic engineering of MbA production. Successful reconstruction of the MbA biosynthetic pathway was achieved in *N. benthamiana* through transient expression of MbA assembly genes together with montbretia flavonol biosynthesis genes [6, 7]. While MbA is the predominant montbretin in montbretia corms, transiently transformed *N. benthamiana* predominantly produced MbB likely due to better availability of coumaroyl-CoA relative to caffeoyl-CoA.

**Fig 1. Structures and inhibitor activity of montbretin A, B and C found in montbretia corms.** (A) Montbretin A (MbA) contains a caffeoyl moiety which contributes to the highly efficient inhibition of human pancreatic α-amylase (HPA). MbB and MbC have a coumaroyl- and feruloyl moiety, respectively, and are not effective HPA inhibitors. (B) $K_i$ values for HPA inhibition were reported previously [4].

The hydroxycinnamoyl-CoA thioesters caffeoyl-CoA, coumaroyl-CoA and feruloyl-CoA are produced from the respective hydroxycinnamic acids by enzymes generally referred to as 4-coumaroyl-CoA ligases (4CLs) [9, 10]. 4CLs belong to the larger family of acyl-activating enzymes (AAEs), which mainly catalyze the conjugation of an organic acid to CoA using an ATP-dependent two-step mechanism [11]. The AAE family can be divided into seven clades [11] with 4CLs clustering in clade IV. However, enzymes possessing 4CL activity have also been identified in clades V, VI, and VII [12–17]. As enzymes in the central phenylpropanoid pathway, 4CLs are ubiquitous across the plant kingdom [18, 19]. They are best known for the production of coumaroyl-CoA which serves as a precursor for a multitude of metabolites, including lignin and flavonoids.

AAE and specifically 4CLs have not been explored in montbretia, where they are likely involved in two different biosynthetic reactions essential for MbA biosynthesis, the production of coumaroyl-CoA for the biosynthesis of the core flavonol myricetin and the production of caffeoyl-CoA as the acyl donor for CcAT in MbA assembly. In this work we explore the molecular basis for the formation of hydroxycinnamoyl-CoA precursors for myricetin and MbA production in montbretia and the utility of Cc4CLs for metabolic engineering. We describe the discovery and characterization of two montbretia 4CLs and their heterologous expression in *N. benthamiana* and *S. cerevisiae* in the context of MbA production.

## Results

### Identification of AAEs expressed in montbretia corms

In order to identify AAEs potentially involved in MbA biosynthesis, we performed a BLASTP search using a set of published AAE sequences from other plant species (S1 File) against a previously described montbretia transcriptome database covering a time course of corm development [6]. We focused on AAEs with high similarity to 4CLs and on AAEs clustering in clades for which enzyme activity with hydroxycinnamic acids has been shown in other species [12–17]. We identified 16 contigs encoding full length montbretia AAEs in clades IV, V, VI, and VII (Fig 2 **and** S2 File). All 16 sequences contained the AMP-binding/adenylate-forming (SSGTTXXPKGV, PROSITE PS00455) motif characteristic for AAEs [11]. A phylogeny of the predicted amino acid sequences revealed two contigs that clustered with known 4CLs (Fig 2). Cc4CL1 (CcAAE2) was most similar to the monocot subclade of class II 4CLs, thought to be involved in phenylpropanoid metabolism other than lignin biosynthesis; while Cc4CL2 (CcAAE10) clustered with monocot class I 4CLs, involved in monolignol biosynthesis [9, 10, 20]. Cc4CL1 and Cc4CL2 share 60% amino acid identity and 79.7% amino acid similarity. Eleven montbretia contigs clustered with clade V, two with clade VI, and one with clade VII (Fig 2). Both clade V and VI AAEs have been shown to contain peroxisomal acyl-CoA ligases, and amino acid sequence analysis revealed that, except for Cc4CL1, Cc4CL2, CcAAE4, CcAAE5, and CcAAE9, all other candidate AAE possess a C-terminal peroxisomal targeting signal (PTS1) (S2 File) [21].

To further narrow down the list of montbretia candidate AAEs, temporal patterns of expression in developing corms were compared to previously characterized MbA biosynthetic genes (MBGs). Transcript abundance in counts-per-million for five developmental time points [6] was used to generate a heatmap and to cluster genes based on expression correlation (Fig 3, S3 **and** S4 Files). We had previously shown that MBGs share a common pattern of transcript expression with highest transcript abundance in the early summer in developing young corms (Fig 3) [5–8]. Based on their expression clustering with MBGs, eight candidate genes *CcAAE1*, *CcAAE3*, *CcAAE4*, *CcAAE5*, *CcAAE7*, *CcAAE9*, and *Cc4CL1* (*AAE2*) and *Cc4CL2* (*AAE10*) were selected for further characterization.

### Cc4CL1 and Cc4CL2 activate hydroxycinnamic acids

The cDNAs covering the full-length open reading frames of the eight AAEs were cloned using cDNA template made from young corm transcripts, and AAEs were individually expressed in *E. coli* for functional characterization. For initial tests for 4CL activity, whole protein extracts were prepared and assayed with ATP, MgCl$_2$, coenzyme A, and individual phenylpropanoid substrates, specifically cinnamic acid, coumaric acid, caffeic acid, ferulic acid, and sinapic acid. A whole protein extract prepared from *E. coli* harbouring an empty vector was used as a negative control. Products were identified by liquid chromatography-mass spectrometry (LC-MS). We observed CoA ligase activity with the above-mentioned substrates only for Cc4CL1 and

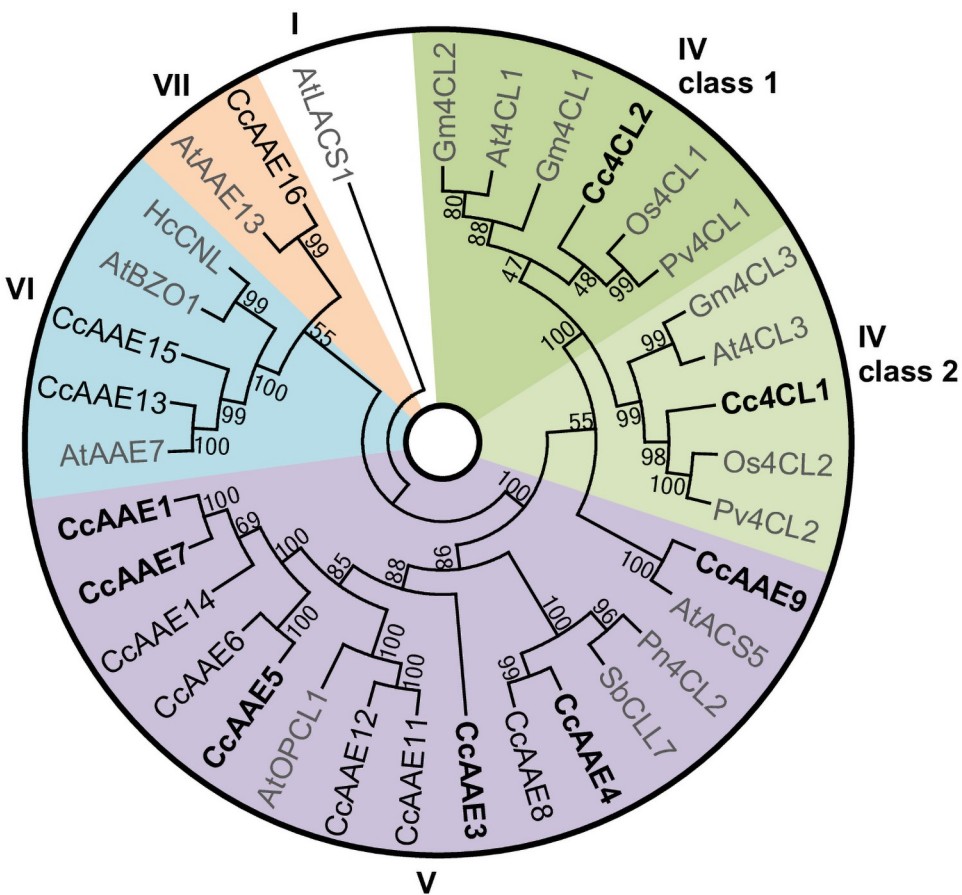

**Fig 2. Phylogeny of montbretia acyl activating enzymes (AAEs).** Predicted amino acid sequences of montbretia AAEs that showed expression patterns that matched those of known MbA biosynthetic genes during young corm development were aligned with selected AAEs from other plant species. A neighbour-joining tree was built and tested with 1000 bootstrap replicates. Bootstrap values are displayed at branch nodes. Enzyme clades I, IV, V, VI, VII are labelled [11]. Montbretia enzymes characterized in this work are in bold. Only AAE clades reported to contain enzymes that activate aromatic acids were considered. AtLACS1 was used as the outgroup. Cc: *Crocosmia x crocosmiiflora*, Gm: *Glycine max*, At: *Arabidopsis thaliana*, Pn: *Piper nigrum*, Os: *Oryza sativa*, Pv: *Panicum virgatum*, Sb: *Scutellaria baicalensis*.

Cc4CL2 (Fig 4 **and** S5 File). Whole protein extracts as well as Ni-purified Cc4CL1 and Cc4CL2 converted coumaric acid, caffeic acid, and ferulic acid to the corresponding CoA thioesters coumaroyl-CoA (peak 1; m/z 912), caffeoyl-CoA (peak 2; m/z 928), and feruloyl-CoA (peak 3; m/z 942) (Fig 4A). In addition, Cc4CL2 also converted cinnamic acid to cinnamoyl-CoA (m/z 896). No activity was detected with sinapic acid. None of the other tested AAEs showed product formation with any of the tested hydroxycinnamic acids. Protein expression was verified by Western blot (S5B File).

Both caffeoyl-CoA and coumaroyl-CoA are required for MbA biosynthesis. In addition, utilization of coumaroyl-, caffeoyl- and feruloyl-CoA by CcAT leads to the formation of MbB, MbA and MbC, respectively. Since Cc4CL1 and Cc4CL2 were both active with caffeic acid and coumaric acid, as well as other hydroxycinnamic acids as substrates, we measured their activity with different substrates *in vitro* with competition assays. Assays were performed with Ni-purified enzyme using equimolar concentrations of the three substrates caffeic acid, coumaric acid and ferulic acid. In competition assays, coumaric acid was the preferred substrate for both

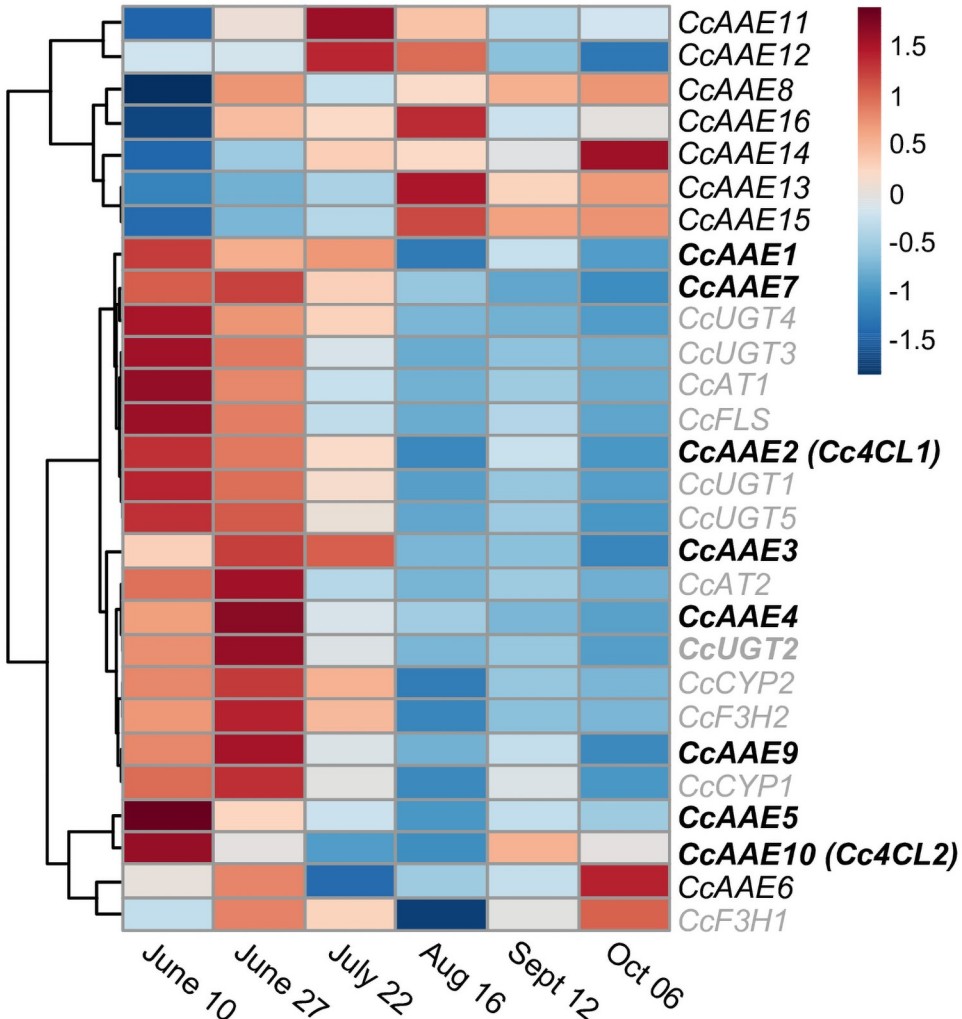

**Fig 3. Temporal expression profile of montbretia AAEs and previously characterized MbA biosynthetic genes.**
Heatmap showing relative transcript abundance of AAEs in montbretia young corms at six different time points of
corm development. Each value represents the mean counts per million of two biological replicates centered and scaled
by transcript. Genes previously characterized as being involved in MbA biosynthesis are shown in gray. Candidate
AAEs selected for characterization in this work are bold. Sampling dates in 2016 are indicated at the bottom.

Cc4CL1 and Cc4CL2 (Fig 4B). Cc4CL1 displayed 4.2 ± 0.7% activity with caffeic acid and no
detectable activity with ferulic acid relative to the activity with coumaric acid (set 100%), while
Cc4CL2 displayed 11.4 ± 0.2% activity with caffeic acid and 26.0 ± 1.8% with ferulic acid rela-
tive to activity with coumaric acid (100%). We observed a lack of stability of the Ni-purified
enzyme which prevented us from assessing enzyme kinetic parameters.

## CcAAEs did not affect MbA levels when transiently expressed in *N. benthamiana*

We previously established a transient *N. benthamiana* montbretin pathway expression system
in which the expression of MBGs enables MbB and trace amounts of MbA formation [6].
Here, we tested the effect of co-expressing *CcAAEs* on montbretin levels. We individually co-
expressed each of the eight candidate CcAAEs, regardless of their *in vitro* 4CL activity, with

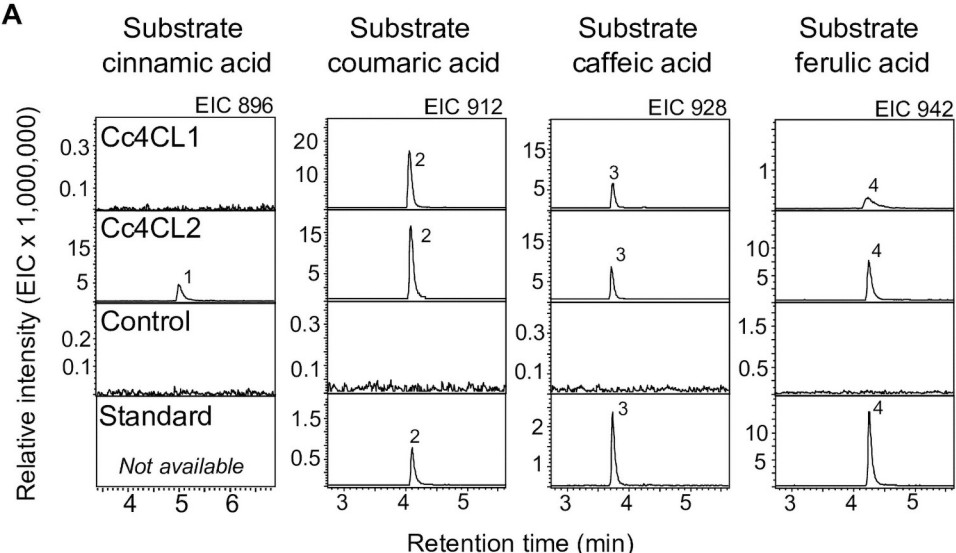

**Fig 4. Enzyme activities of Cc4CL1 and Cc4CL2.** (A) LC-MS product profiles. Enzymes were heterologously expressed in *E. coli* and Ni-purified protein was assayed with ATP, MgCl$_2$, CoA, and one of the hydroxycinnamic acids, cinnamic acid, coumaric acid, caffeic acid, or ferulic acid. As controls, assays were performed with purified protein extracts of *E. coli* transformed with the empty vector. Products were analyzed using LC-MS and extracted ion chromatograms (EIC) are shown. Peak 1, tentatively identified as cinnamoyl-CoA (m/z 896); peak 2, coumaroyl-CoA (m/z 912); peak 3, caffeoyl-CoA (m/z 928); peak 4, feruloyl-CoA (m/z 942). n.d. not detected. (B) Relative product formation in competition assays. Cc4CL1 or Cc4CL2 were assayed with a combined set of the three hydroxycinnamic acid substrates, coumaric acid, caffeic acid and ferulic acid, in one reaction. The 100% activity value was defined by the substrate that yielded the most abundant product in the competition assays.

the MBGs (35S$_{pro}$:(*CcMYB4* + *CcFLS* + *CcCYP2* (CYP75B138) + *CcUGT1* + *CcUGT2* + *CcUGT3* + *CcUGT4* + *CcUGT5* + *CcAT1*) + pBIN:*p19*) (S6 File) in *N. benthamiana* leaves. Expression of *Cc4CL1* or *Cc4CL2* in combination with MBGs did not affect montbretin product profiles relative to expression of MBGs alone (Fig 5 **and** S7 **and** S8 Files). MbB was the major product formed (371.4 ± 31.3 μg/g), followed by MbC (79.9 ± 10.6 μg/g), and lower amounts of MbA (6.3 ± 1.4 μg/g). Beyond Cc4CL1 and Cc4CL2, expression of the other CcAAEs also had no effect on the levels of MbA ($F_{8,18}$ = 1.566, p = 0.204), MbB ($F_{8,18}$ = 1.426, p = 0.252) or MbC ($F_{8,18}$ = 1.064, p = 0.429) (S7 **and** S8 Files).

## Utilizing Cc4CLs to reconstruct flavanone biosynthesis in *Saccharomyces cerevisiae*

Unlike plants, *S. cerevisiae* does not have endogenous 4CL activity which is essential for flavonoid biosynthesis. We tested the utility of *Cc4CL1* and *Cc4CL2* in *S. cerevisiae* to convert exogenously fed *p*-coumaric acid to *p*-coumaroyl-CoA, and exogenously fed caffeic acid to caffeoyl-CoA, to support formation of the flavanones naringenin and eriodictyol (Fig 6). Naringenin and eriodictyol are precursors for myricetin, which is the core building block of MbA (Fig 1). For metabolic engineering of montbretin in yeast, 4CL activity would also enable the

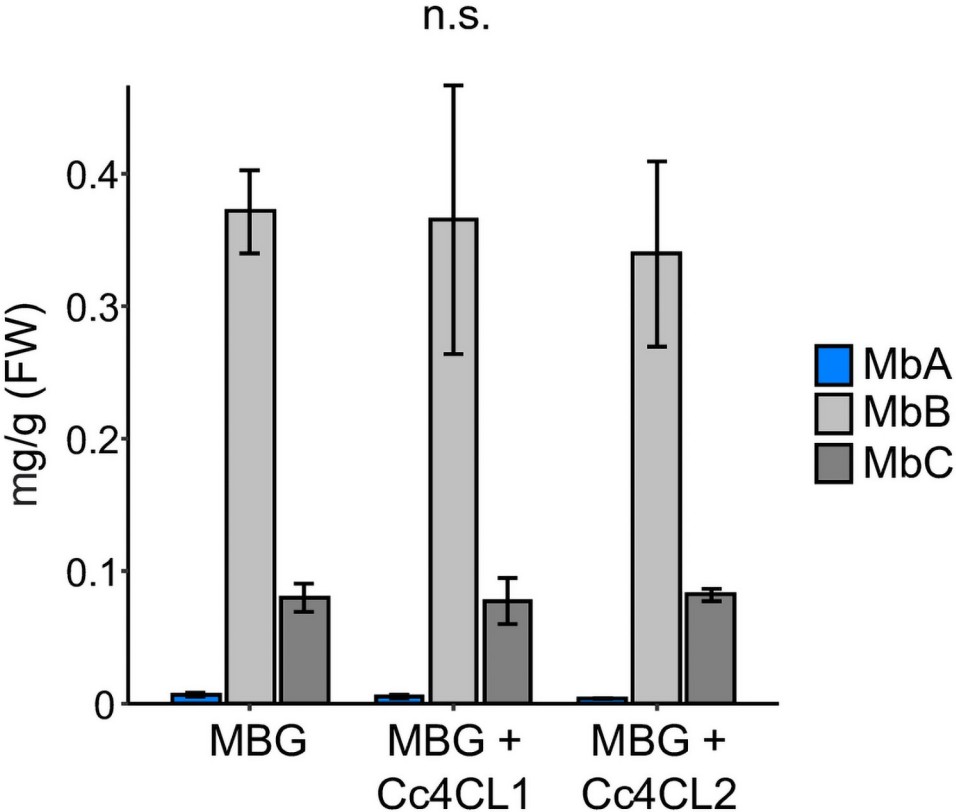

**Fig 5. Montbretin accumulation in *N. benthamiana* co-expressing MbA biosynthetic genes and Cc4CL1 or Cc4CL2.** *N. benthamiana* leaves were infiltrated with different combinations of *A. tumefaciens* transformed with plasmids carrying the *35S*-promoter-gene constructs for myricetin and MbA biosynthetic genes (= MBGs, [6]) as well as *4CL1* or *4CL2* (see S6 File). Leaves were collected 5 days post-infiltration. Metabolites were extracted with 50% MeOH, analyzed by LC-MS. Data represents mean ± SE (n = 4). MbA ($F_{8,18}$ = 1.57, p = 0.204), MbB ($F_{8,18}$ = 1.43, p = 0.25), MbC ($F_{8,18}$ = 1.06, p = 0.43), n.s. not significant.

formation of *p*-coumaroyl-CoA or caffeoyl-CoA as acyl donors for CcAT in MbB or MbA biosynthesis, respectively.

We expressed *Cc4CL1* or *Cc4CL2* together with the previously described *CcCHS2* and *CcCHI2* in *S. cerevisiae* [7]. Constructs were assembled using the yeast toolkit for modular cloning (MoClo-YTK) [22] to express all three genes from one plasmid using constitutive promotors of different strengths (Fig 6A **and** S9 File). Plasmids were transformed into *S. cerevisiae* BY4741, cultures were fed with *p*-coumaric acid or caffeic acid and grown in 48-well plates for 67 h. Naringenin or eriodictyol formation was detected using LC-MS. Both 4CLs activated *p*-coumaric or caffeic acid to provide hydroxycinnamoyl-CoA thioesters necessary for flavanone production as shown by the formation of naringenin or eriodictyol, respectively (Fig 6C and 6D). Hydroxycoumaroyl triacetic lactone formation occurred as a side product (S10 File). Expression of Cc4CL1 resulted in slightly higher levels of naringenin production (5.8 mg $^*$ L$^{-1}$) when *p*-coumaric acid fed, compared to Cc4CL2 (5.2 mg $^*$ L$^{-1}$); eriodictyol production (caffeic acid fed) was higher with the expression of Cc4CL2 (2.3 mg $^*$ L$^{-1}$) compared to Cc4CL1 (1.5 mg $^*$ L$^{-1}$). Naringenin titers were about 3-fold higher compared to eriodictyol titers. The formation of the side product, triacetic acid lactone was more pronounced when caffeic acid was fed and eriodictyol produced. No product formation could be detected using a GFP control vector or when no hydroxycinnamic acid was fed.

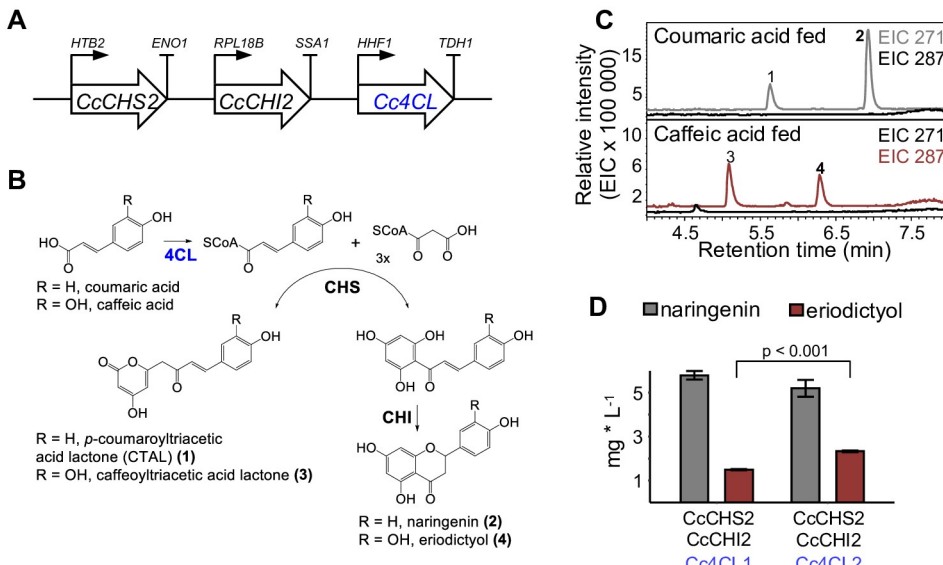

**Fig 6. Reconstruction of flavanone biosynthesis in *Saccharomyces cerevisiae* using montbretia 4CLs.** (A) Flavanone biosynthesis module. For expression in *S. cerevisiae*, expression modules containing montbretia chalcone synthase (*CcCHS2*), chalcone isomerase (*CcCHI2*) and one of two 4-coumaroyl CoA ligases (4CLs), montbretia *Cc4CL1* or *Cc4CL2*, were constructed. (B) Schematic representation of flavanone biosynthetic pathway. Coumaric acid or caffeic acid are converted into naringenin or eriodictyol, respectively. Lactone formation occurs as a side reaction. (C,D) Engineering flavanone production in *S. cerevisiae*. Flavanone biosynthesis modules (see A) differing in their Cc4CL genes were heterologously expressed in *S. cerevisiae*. Genes were constitutively expressed from the same low copy plasmid. Yeast cultures were grown for 67h in 48-well plates, extracted and analysed for flavanone formation using LC-MS. Representative extracted ion chromatograms (EIC) (C) and amounts of produced naringenin or eriodictyol are shown (n = 2) (D). Peak 1, tentatively identified as *p*-coumaroyltriacetic acid lactone; peak 2, naringenin; peak 3, tentatively identified as caffeoyltriacetic acid lactone; peak 4, eriodictyol. Changes in flavanone levels were tested using Welch's two-sample t-tests (naringenin: p = 0.27; eriodictyol: p < 0.001).

## Discussion

Previous research on the biosynthesis of the anti-diabetic and anti-obesity compound MbA has focused mostly on the unique assembly of MbA from a suite of common plant metabolites (i.e., MbA building blocks), specifically the flavonoid myricetin, the phenylpropanoid caffeoyl-CoA, and the UDP-sugars UDP-rhamnose, UDP-glucose and UDP-xylose [5–8]. The central phenylpropanoid pathway [23] produces the precursor coumaroyl-CoA for myricetin, and it also produces caffeoyl-CoA, thus feeding twice into the MbA biosynthetic system. While caffeoyl-CoA is required for the acylation reaction in the MbA assembly pathway, the similar phenylpropanoids coumaroyl-CoA and feruloyl CoA provide the acyl functionalities for MbB and MbC, respectively (Fig 1). MbB and MbC have no significant potential for treatment of diabetes or obesity [2], but their biosynthesis in montbretia or in metabolically engineered tobacco or yeast competes with MbA production [2, 7]. Here, we investigated montbretia AAEs with a focus on 4CLs and their ability to convert caffeic acid, coumaric acid or ferulic acid into caffeoyl-CoA, coumaroyl-CoA or feruloyl-CoA, respectively. We identified two Cc4CLs, both of which showed activity with all three hydroxycinnamic acids and an apparent preference for coumaric acid as substrate relative to caffeic acid and ferulic acid. While expression of AAEs failed to enhance MbA biosynthesis in tobacco, we demonstrated that both Cc4CL1 and Cc4CL2 can be used to activate coumaric and caffeic acid towards flavanone biosynthesis in metabolically engineered yeast.

## Cc4CL1 and Cc4CL2 are not substrate specific

The CcATs that catalyze the formation of montbretins can use caffeoyl-CoA, coumaroyl-CoA and feruloyl-CoA, thus leading to the formation of MbA, MbB and MbC [5]. Given the lack of CcAT substrate specificity it is possible that preferential biosynthesis of any one of these three montbretins might be controlled by differences in the availability of caffeoyl-CoA, coumaroyl-CoA or feruloyl-CoA. In addition, all three montbretins require coumaroyl-CoA for the myricetin core. We hypothesized that montbretia corms might express a set of different Cc4CLs with distinct substrate specificities as a means to regulate the formation of caffeoyl-CoA, coumaroyl-CoA and feruloyl-CoA. However, the results of this work indicate that this is not the case. Substrate promiscuity of 4CLs has also been observed in other plant species [24–28]. The majority of 4CLs have been studied in dicot species where they generally display a preference for *p*-coumaric acid or caffeic acid [24, 29, 30]. Fewer 4CLs have been characterized in monocots. In rice (*Oryza sativa*), switchgrass (*Panicum virgatum*), and great millet (*Sorghum bicolor*), 4CLs generally prefer coumaric acid and ferulic acid relative to caffeic acid [25, 31, 32]. This pattern holds true for Cc4CL2, which displayed preference for *p*-coumaric acid and ferulic acid compared to caffeic acid, while Cc4CL1 did not accept ferulic acid in competition assays.

In addition to *bona fide* 4CLs, some AAEs outside the 4CL clade have been reported to use phenylpropanoid substrates. For example, AAEs in clades V, VI, and VII contribute to benzenoid metabolism through activation of cinnamic acid [12, 13, 33, 34]. In yew (*Taxus media*) AAEs outside of the 4CL clade can activate hydroxycinnamic acids with TmAAE12 preferring caffeic acid *in vitro* [17]. Arabidopsis (*Arabidopsis thaliana*) clade V AAEs also activate *p*-coumaric acid and/or caffeic acid [33, 35]. In contrast, none of the CcAAEs outside of the 4CL clade showed activity with hydroxycinnamic acids. Thus, 4CL activity in montbretia appears to be limited to the *bona fide* 4CL clade, and does not contribute to selective biosynthesis of MbA relative to other montbretins.

## Cc4CL1 is likely involved in MbA biosynthesis

4CLs cluster within clade IV of the AAE family. Within clade IV, two functionally distinct sub-clades exist, class I and class II. Enzymes involved in lignin biosynthesis cluster within class I, while class II contains 4CLs that are likely involved in soluble phenylpropanoid metabolism, including flavonoid biosynthesis [9, 10, 20]. Cc4CL1 clusters within class II 4CLs, accepts coumaric acid and caffeic acid as substrates, and its gene expression pattern matches that of MBGs during young corm development. Cc4CL2 clusters within class I, accepts coumaric acid, caffeic acid as well as ferulic acid as substrates, but its expression profile pattern does not match that of other MBGs. Both Cc4CL1 and Cc4CL2 show high transcript abundance in early summer. These observations suggest that Cc4CL1 is more likely to contribute to MbA biosynthesis while Cc4CL2 may be involved in lignin biosynthesis.

## CcAAEs do not enhance MbA production in *N. benthamiana*

We previously engineered *N. benthamiana* for the production of MbA by transiently co-expressing a suite of genes for myricetin and MbA biosynthesis [6, 7]. This system yielded small amounts of less than 10 μg MbA per g fresh weight of tobacco leaf material, while levels of MbB and MbC were about 60-fold and 15-fold, respectively, higher. We showed here that co-expression of AAEs, including Cc4CL1 and Cc4CL2, did not alter these product profiles suggesting that 4CL activity is not rate limiting for overall montbretin production in *N. benthamiana*. However, future work shall focus on the possibility to enhance caffeoyl-CoA formation in *N. benthamiana* in order to increase the relative ratio and overall yield of MbA. This

may be achieved through selection or mutagenesis of a 4CL enzyme with higher substrate preference for caffeic acid. The *Arabidopsis thaliana* At4CL2 was shown to have higher activity toward caffeic acid [29] and could also be explored for increasing caffeoyl-CoA formation in *N. benthamiana*.

### Engineering flavanone biosynthesis in *Saccharomyces cerevisiae*

*N. benthamina* and yeast each have unique advantages and also provide challenges for metabolic engineering of MbA production. In contrast to *N. benthamiana*, which naturally produces all of the MbA building blocks, MbA biosynthesis in yeast will require more extensive engineering for most of the MbA building blocks, in addition to engineering of the MbA assembly pathway. However, the fact that yeast does not produce caffeic acid/caffeoyl-CoA and coumaric acid/coumaroyl-CoA, may allow for better control of MbA versus MbB production in yeast by controlled feeding of caffeic acid. CHS which is the entry enzyme for flavonol biosynthesis efficiently uses caffeic acid although coumaric acid is generally the preferred substrate [36]. Here we demonstrated the feasibility of producing the flavanones eriodictyol and naringenin, two possible precursors to myricetin, from caffeic acid and coumaric acid, respectively, by heterologous co-expression of Cc4CLs with montbretia flavanone biosynthetic genes. The yields of these flavanones were comparable to eriodictyol and naringenin production reported for yeast ectopically expressing flavanone biosynthetic genes, while higher levels can be achieved through genomic integration and pathway optimization [37–40]. In future work we will further optimize eriodictyol production from caffeic acid in yeast, similar to the previously described naringenin production [39, 40].

## Conclusion

The caffeoyl moiety is essential for the anti-diabetic function of MbA. Here we investigated the production of hydroxycinnamic CoA thioesters for myricetin and MbA production in montbretia. Montbretia expresses two functional Cc4CLs, Cc4CL1 and Cc4CL2. Cc4CL1 is likely involved in MbA biosynthesis, while Cc4CL2 may contribute to lignin biosynthesis. Cc4CL activity does not enhance MbA production or the production of other montbretins, MbB and MbC, when expressed together with MbA pathway genes in *N. benthamiana*. Both Cc4CLs can be used for flavanone production in yeast to support metabolic engineering of MbA.

## Materials and methods

### Identification of target CcAAE

RNA extraction from montbretia corms, cDNA synthesis, and corm time-course transcriptome sequencing, assembly, and differential expression analysis was described previously [5, 6]. To identify putative AAEs involved in MbA biosynthesis, a BLASTP search of the translated montbretia transcriptome was conducted using a set of characterized AAE (S1 File). Results were filtered using a reciprocal BLASTP search against the non-redundant protein database of NCBI. Resulting montbretia AAEs were used in a second reciprocal BLASTP analysis. Only sequences clustering into clade IV, V, VI and VII of the AAE family were considered as candidates. This process resulted in 38 putative AAEs. Sixteen of those were longer than 400 amino acids and retained for further analysis.

Transcript expression data in counts per million (cpm), from six different developmental time-points of montbretia young corms, were generated as previously described [6]. We used this time-course expression data to compute Pearson correlation coefficients (S3 File). Genes

which clustered into the 4CL clade and/or showed a correlation coefficient $> 0.9$ were selected yielding eight final candidate genes. Expression data for 16 AAE can be found in S4 File.

A heatmap visualizing expression patterns of putative CcAAEs alongside characterized montbretia myricetin and MbA biosynthetic genes was generated in R 4.0.2 [41] using the 'pheatmap' package [42]. Genes possessing similar transcript expression patterns were determined by hierarchical clustering using Pearson correlation distance and complete-linkage clustering. Each data point represents normalized mean counts per million from two biological replicates.

The predicted amino acid sequences of putative montbretia AAEs and AAE from other plants were aligned using ClustalW algorithm and the GONNET protein weight matrix implemented in MEGA7 [43] (S1 File). This alignment was used to generate a phylogenetic tree with MEGA7 using the neighbour-joining algorithm and the Poisson model. The tree topology was evaluated by performing a bootstrap resampling analysis with 1000 replicates.

## Cloning and heterologous expression in *E. coli*

The eight selected *CcAAE*s were amplified from young corm cDNA (harvested June $10^{th}$, 2016) and cloned into the pJet1.2/blunt vector for sequencing (S11 File) [5]. Complete open reading frames of the target *CcAAE*s were cloned as *Bsa*I *or Bbs*I fragments into the pAS-K-IBA37+ expression vector (IBA), for N-terminal His$_6$-tagging. The *E. coli* TOP10 strain (Invitrogen) was used for heterologous *CcAAE* expression. Cultures were grown at 21˚C, 220 rpm and induced at $OD_{600} = 0.6$ with 200 μg $^*$ L$^{-1}$ anhydrotetracycline (Sigma-Aldrich) and further cultured at 18˚C for 20 h. Cultures were collected by centrifugation (4300 x $g$) and disrupted by five freeze and thaw cycles in chilled extraction buffer (50 mM Tris-HCl, pH 7.5, 10 mM MgCl$_2$, 5 mM DTT, 10% [v/v] glycerol, 1x Pierce™ protease inhibitor [EDTA-free; Thermo Fisher Scientific], 25 U Benzonase Nuclease [Merck, Germany], and 0.2 mg $^*$ mL$^{-1}$ lysozyme). Cell fragments were removed by centrifugation (14 000 x $g$), and the supernatant was desalted into assay buffer (10 mM Tris-HCl, pH 7.5, 1 mM DTT, and 10% [v/v] glycerol) using Econopac 10DG columns (Bio-Rad, USA). For protein purification of 4CL1 and 4CL2 using Ni-NTA, a modified extraction buffer was used (50mM Tris-HCl, pH 7.5; 10mM MgCl$_2$;5mM DTT; 2% [v/v] glycerol; 150 mM NaCl$_2$; 20 mM imidazole; 1x Pierce protease inhibitor [EDTA-free, ThermoFisher Scientific], 25 U Benzonase Nuclease [Merck, Germany], and 0.2 mg $^*$ mL$^{-1}$ lysozyme) and the lysate was directly loaded onto a Ni-NTA agarose column (Qiagen). Protein was eluted with elution buffer (10 mM Tris-HCl, pH 7.5; 400 mM imidazole; 1mM DTT; 10% [v/v] glycerol) and desalted into assay buffer using Illustra NAP-5 Columns (GE Healthcare). Enzyme assays were set up the same day as storage at 4˚C or shock-freezing of the purified protein led to loss of activity. The Monoclonal AntipolyHistidine-Alkaline Phosphatase antibody (Sigma-Aldrich) and the 1-Step NBT/BCIP Substrate Solution (Thermo Fisher Scientific) were used to ensure successful heterologous enzyme expression (S5B File).

## Enzyme assays with recombinant CcAAEs

To test for 4CL activity, initial enzyme assays were performed with 100 μL bacterial extract, 5 mM ATP, 5 mM MgCl$_2$, 200 μM CoA (Sigma-Aldrich, Germany), and 100 μM of substrate in a Teflon-sealed, screw-capped 1 mL glass vial for 2 h at 25˚C. Substrates tested were cinnamic acid, coumaric acid, caffeic acid, ferulic acid, and sinapic acid (Sigma-Aldrich). For the competitive assays, Ni-purified enzyme was incubated with 100 μM of each substrate, caffeic acid, coumaric acid and ferulic acid, at 25˚C for 10, 20 or 30 minutes (two replicates each). Assays were stopped by adding 100 μL methanol and placing on ice. Products were detected by liquid

chromatography-mass spectrometry (LC-MS). The identity of coumaroyl-CoA, caffeoyl-CoA, and feruloyl-CoA was confirmed by comparing mass and retention time with authentic standards. For competitive assays, peak area of extracted ion chromatograms was used to calculate percentage of products formed.

### Transient expression of CcAAEs in *Nicotiana benthamiana*

For transient expression in *N. benthamiana*, the coding sequence of the eight *CcAAEs* were separately cloned into the pCAMBiA2300U vector. After sequence verification, pCAMBiA vectors carrying the *CcAAEs* as well as pCAMBiA vectors carrying the previously described genes for myricetin biosynthesis *CcFLS*, *CcCYP2* and *CcMYB4* [7] and the genes for MbA biosynthesis *CcUGT1*, *CcUGT2*, *CcAT1*, *CcUGT3*, *CcUGT4* and *CcUGT5* (all nine genes are referred to as montbretin biosynthetic genes = MBGs) [5, 6, 8] and the pBIN:*p19* were individually transformed into *Agrobacterium tumefaciens* strain C58pMP90. Agrobacterium cultures were prepared as previously described [6]. Briefly, overnight cultures were grown (220 rpm, 28 ˚C) and used to inoculate 10 mL LB-media containing 50 μg $^{*}$ mL$^{-1}$ kanamycin, 25 μg $^{*}$ mL$^{-1}$ rifampicin and 25 μg $^{*}$ mL$^{-1}$ gentamicin for overnight growth. The following day the cultures were centrifuged (4,000 x *g*, 5 min) and cells were re-suspended in infiltration buffer (10 mM MES, 10 mM MgCl$_2$, 100 μM acetosyringone, pH 5.6) to a final OD$_{600}$ of 0.5. After shaking for 2 h at 21˚C, combinations were prepared for leaf infiltration by mixing equal volumes of transformed *A. tumefaciens*. MBGs were used as a control: *A. tumefaciens* 35S$_{pro}$:(*CcMYB4* + *CcFLS* + *CcCYP2* + *CcUGT1* + *CcUGT2* + *CcUGT3* + *CcUGT4* + *CcUGT5* + *CcAT1*) + *A. tumefaciens* pBIN:*p19*. AAE were individually tested in the background of MBGs: MBGs + *A. tumefaciens* 35S$_{pro}$:(*CcAAE*) (AAE being 1/11 of the mixture; S6 File). The leaves of four-week-old *N. benthamiana* plants were infiltrated with these *Agrobacterium* suspensions using a 1-mL needleless syringe to gently push the suspension into the leaf's abaxial surface. Infiltrated leaves were labeled with tape and harvested five days after infiltration. Leaf discs 1 cm in diameter were cut from harvested leaves using a cork borer. Leaf discs were then weighed, and metabolites were extracted using 1 mL of 50% MeOH (v/v) per 100 mg of tissue. Extracts were analyzed using LC-MS.

### Reconstruction of flavanone biosynthesis in yeast

For heterologous expression in yeast the yeast toolkit for modular cloning (MoClo-YTK) was used (Addgene, MA, USA) [22]. First, the montbretia flavanone biosynthetic genes, *CcCHS2*, *CcCHI2* [7], and *Cc4CL1*, *Cc4CL2* (this work), were individually cloned into the entry vector, sequenced, paired with a promotor and terminator (Fig 6 **and** S9 File) and cloned into a transcriptional unit. Transcriptional units were then combined to multigene constructs containing CcCHS2, CcCHI2 and one of the 4CLs; *Cc4CL1* or *Cc4CL2* (S9 File). This yielded two flavanone biosynthesis expression modules, each differing in their *4CL*. Details about promotors, terminators and parts for the vector backbone can be found in S9 File. Each module was individually transformed into the yeast strain BY4741 (GE Life Sciences, http://www. gelifesciences.com/). For flavanone biosynthesis, a single yeast colony was used to inoculate a starting culture in 5 mL selective uracil dropout media (SD-URA), which was grown overnight at 28 ˚C and 180 rpm. The following day, 250 μL SD-URA media were inoculated to an OD of 0.1, fed with coumaric or caffeic acid (100 μM end concentration; 2.5 μL of 100 mM stock dissolved in ethanol) and grown in 48-well plates covered with a sterile-breathable sealing film (VWR, ON, Canada) for 67h at 28 ˚C and 180 rpm. To prevent evaporation, plates were placed in a plastic container under high humidity. Three replicates were grown for each sample and the experiment was repeated two times independently. For metabolite analysis, 50 μL sample

was mixed with 50 μL methanol and the supernatant was analysed as described in the LC-MS section.

## LC-MS analyses

LC analysis was done on an Agilent 1100 HPLC (Agilent Technologies, Waldbronn, Germany) with Agilent ZORBAX SB-C18 column (50 x 4.6 mm, 1.8 μm particle size) (Merck, Darmstadt, 370 Germany). *In vitro* enzyme assays were analyzed using 10 mM aqueous ammonium acetate (Mobile phase A) and acetonitrile plus formic acid (0.2% v/v; Mobile phase B) with the following elution profile: 0–5 minutes, 5–35% B in A; 5 to 9 minutes, 35–90% B in A. The flow rate was 0.8 mL x min$^{-1}$ at a column temperature of 50 ˚C. *N. benthamiana* extracts and yeast assays were analyzed using aqueous formic acid (0.2% v/v; Mobile phase A) and acetonitrile plus formic acid (0.2% v/v; Mobile phase B). The elution profile for *N. benthamiana* leaf extracts was: 0–0.5 min, 95% A; 0.5–5 min, 5–20% B in A; 5–7 min 90% B in A and 7.1–10 min 95% A. The flow rate was 0.8 mL x min$^{-1}$ at a column temperature of 50 ˚C. The elution profile for flavanones produced in yeast was: 0–0.5 min, 95% A; 0.5–5 min, 5–40% B in A; 5–7 min 90% B in A and 7.1–10 min 95% A. The flow rate was 0.8 mL x min$^{-1}$ at a column temperature of 50 ˚C.

LC was coupled to an Agilent MSD Trap XCT-Plus mass spectrometer using electrospray ionization functioning in negative mode (capillary voltage, 4000 eV; temp, 350 ˚C; nebulizing gas, 60 psi; dry gas 12 L/min). MS/MS was used to monitor daughter ion formation. Agilent LC/MSD Trap Software 5.2 (Bruker Daltonik) was used for data acquisition and analysis. MbA, MbB, and MbC production in *N. benthamiana* was quantified using an external MbA standard curve and flavanone production in yeast was quantified using naringenin or eriodictyol standard curves. Compounds were identified using their retention times, molecular masses, and specific fragmentation patterns and by use of authentic standards if they were available (eriodictyol and naringenin, Sigma-Aldrich; MbA and MbB, [2, 5]).

## Statistical analysis

One-way analysis of variance (ANOVA) was used to test for significant differences in the accumulation of MbA, MbB and MbC in *N. benthamiana*. Welch's two-sample t-test was used to test for differences in flavanone production in yeast. These analyses were conducted in base R 4.0.2 [41].

## Accession numbers

Nucleotide sequences were deposited in GenBank with the accession numbers MZ944850 (Cc4CL1), MZ944849 (Cc4CL2). Accession numbers or sequences for all other genes/proteins used in this work are indicated in S1 **and** S2 Files. Previously published transcriptome libraries as well as time course data described in the manuscript are available in the NCBI/GenBank Sequence Read Archive (SRA) under the project PRJNA389589 (SRP108844).

## Supporting information

**S1 File. Protein accession numbers.**
(DOCX)

**S2 File. Amino acid sequences of candidate CcAAEs.**
(XLSX)

**S3 File. Correlation of *CcAAEs* expression with previously identified MbA pathway genes during young corm development.**
(DOCX)

**S4 File. Expression data of 16 candidate *AAEs*.**
(XLSX)

**S5 File. Activity and expression of *AAEs*.**
(DOCX)

**S6 File. Combinations of genes used for transient expression in *N. benthamiana*.**
(DOCX)

**S7 File. Montbretin A, B and C levels in *N. benthamiana* plants transiently expressing MBGs and AAE.**
(DOCX)

**S8 File. Identification of montbretins produced by *N. benthamiana* transiently expressing MBGs.**
(DOCX)

**S9 File. Parts of the MoClo system used for the construction of *S. cerevisiae* expression modules.**
(DOCX)

**S10 File. Identification of products formed by *S. cerevisiae* expressing a flavanone biosynthetic module.**
(DOCX)

**S11 File. Oligonucleotides used in this study.**
(DOCX)

## Acknowledgments

We thank Stephen G. Withers for insightful discussions and collaboration, Carol Ritland for project management and Macaire M. S. Yuen for bioinformatic support.

## Author Contributions

**Conceptualization:** Joerg Bohlmann, Sandra Irmisch.

**Formal analysis:** Frederick G. Sunstrum, Sharon Jancsik, Sandra Irmisch.

**Funding acquisition:** Joerg Bohlmann.

**Investigation:** Frederick G. Sunstrum, Hannah L. Liu, Sharon Jancsik, Lufiani L. Madilao, Joerg Bohlmann, Sandra Irmisch.

**Methodology:** Sharon Jancsik.

**Resources:** Joerg Bohlmann.

**Supervision:** Joerg Bohlmann, Sandra Irmisch.

**Writing – original draft:** Frederick G. Sunstrum, Joerg Bohlmann, Sandra Irmisch.

**Writing – review & editing:** Sharon Jancsik, Joerg Bohlmann, Sandra Irmisch.

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
