## [Decision Letter · Decision Letter 0]

26 Aug 2021

PONE-D-21-23997

4-Coumaroyl-CoA Ligases Involved in the Biosynthesis of the Anti-Diabetic Metabolite Montbretin A

PLOS ONE

Dear Dr. Bohlmann,

Thank you for submitting your manuscript to PLOS ONE. After the evaluation of two experts in the field, we consider that your article can be accepted for publication after you have carried out the corrections suggested by the reviewers. Therefore, we invite you to submit a revised version of the manuscript that addresses the points raised during the review process.

Note that, although the opinions from the reviewers were quite positive, they did raised interesting points to improve the manuscript. Some changes might be considered and additional information might be added.

We look forward to receiving your revised manuscript.

Kind regards,

Igor Cesarino, Ph.D

Academic Editor

PLOS ONE

Journal Requirements:

Reviewers' comments:

Reviewer's Responses to Questions

**Comments to the Author**

1. Is the manuscript technically sound, and do the data support the conclusions?

Reviewer #1: Yes

Reviewer #2: Yes

2. Has the statistical analysis been performed appropriately and rigorously? 

Reviewer #1: Yes

Reviewer #2: Yes

3. Have the authors made all data underlying the findings in their manuscript fully available?

Reviewer #1: Yes

Reviewer #2: No

4. Is the manuscript presented in an intelligible fashion and written in standard English?

Reviewer #1: Yes

Reviewer #2: Yes

5. Review Comments to the Author

Reviewer #1: PONE-D-21-23997, 4-Coumaroyl-CoA Ligases Involved in the Biosynthesis of the Anti-Diabetic Metabolite Montbretin A

Sunstrom and co-workers approach an interesting problem in the engineered biosynthesis of the potential therapeutic plant specialized metabolite Montbretin A. It was previously observed that it accumulates only as a minor part in a mixture of related, but unwanted compounds when functionally characterized steps of the pathway were heterologously expressed in the transient Nicotiana benthamiana system. Here, a family of 4-coumarate CoA ligases is identified in Crocosmia, where Montbretin A is a major metabolite.

These enzymes have evolved to provide precursors for specific phenylpropanoid pathways but are known to display significant functional redundancy as well. Solid biochemical characterization of the enzymes demonstrated relevant activities for two of the family toward phenylpropanoid substrates, including those involved in Montbretin biosynthesis. However, this also showed functional redundancy and non-selectivity for their substrates. Engineering of the pathway in N. benthamiana, including these novel enzymes, did not show the hoped-for shift in the distribution of the formed product pattern.

Deployment of the enzymes in yeast supports and validates their function.

Discussion of the results is comprehensive and possible future perspectives, i.e., the use of recombinant enzymes outside of Cocosmia or the engineering of the substrate specificity are valid points for follow-up.

Despite these seemingly negative results, this study remains solid and in my opinion is well suited for PlosOne’s publication criteria. Below are minor comments, or suggestions.

It is possible that the product pattern in Crocosmia is result of a distinct substrate pattern, i.e., much higher concentration of caffeic acid. What are the specific expression pattern of HCT, Hydroxycinnamoyl CoA shikimate:quinate hydroxycinnamoyltransferase, C3′H, p-coumaroyl shikimate 3′-hydroxylase and CSE, caffeoyl shikimate esterase? These are possibly driving the caffeate formation.

The Arabidopsis 4CL2 isoform was shown to have higher activity toward caffeate and was also investigated for its substrate specificity, albeit against sinapate.

I strongly suggest to have cytochromes P450 enzymes annotated by David Nelson, instead of using a generic nomenclature.

L 193, symbol font

L 472, font color

Phylogeny, I suggest collapsing branches with less than 50% bootstrap support.

Fig. 6, I suggest labelling the tentatively identified products in panel C in the schematic representation B.

Accession numbers are missing.

Reviewer #2: The manuscript by Sunstrum et al provides a molecular characterization of a set of acyl-activating enzymes focusing on 4-coumarate-CoA ligases (4CL) from Montbretia. Two putative 4CLs were identified through sequence comparison and phylogenetic reconstruction as likely 4CLs and were biochemically confirmed as bona fide 4CLs through heterologous expression in E. coli and end-point enzyme activity tests. Both 4CLs showed substrate utilization profiles expected for 4CLs. An additional six related, yet distinct acyl-activating enzymes showed no activity with the ‘standard’ range of 4CL substrates. No additional substrates were tested.

The rational for characterizing the 4CL family from Montbretia was previous work on the biosynthesis and metabolic engineering of a species-specific set of secondary metabolites called montbretins (MbA, MbB, and MbC). Only MbA has medicinal activity as a potential anti-diabetic, while other montbretins do not. Heterologous expression of montbretin biosynthetic genes in Nicotiana yielded MbB and MbC production but only trace amounts of the desired MbA. The bioactive MbA requires caffeoyl-CoA as a precursor, while other montbretins require other 4CL products. The hypothesis was that Montbretia possesses a caffeate-specific 4CL isoform specific for MbA biosynthesis that could increase MbA yield also in heterologous hosts such as yeast or Nicotiana. This hypothesis is intriguing, but proved to be wrong as none of the tested enzymes was specific for caffeic acid (instead were ‘classical’ 4CLs able to utilize multiple substrates) nor did they increase MbA (or MbB or MbC) production in Nicotiana. The latter suggests that 4CL-generated precursors are not limiting heterologous production of montbretins in Nicotiana.

Towards establishing a MbA biosynthetic pathway in yeast, the authors co-expressed the Montbretia 4CLs with two initial steps of flavonoid biosynthesis and showed that Montbretia genes can reconstitute the initial steps of the pathway in yeast, confirming other approaches using the corresponding genes from other plants.

Overall, the manuscript is extremely well written, clearly describes all experimental designs and rationales and provides sufficient details on the methods employed. It is rare that I do not have any editorial suggestions to improve a manuscript, but this is the case here.

Hera are a few minor points for discussion and possible improvement:

The title may be perceived as misleading, because no functional (e.g. reverse genetic) data are presented that prove involvement of the 4CLs in montbretin biosynthesis. It may also be an overstatement that MbA is an anti-diabetic; without clinical studies are published, it should rather be considered a candidate.

The supporting information show a Western lot highlighting successful heterologous expression of the enzymes tested. It is unclear what antibody has been used (anti-His?) and if protein crude extracts have been used or purified proteins as the method is not described in the M&M section. It is noteworthy that 4CL1 and 4CL2 had highest expression levels (and the largest amounts of unspecific labelling); is it just a coincidence that the only two active enzymes also expressed best?

NCBI accession numbers for the 4CLs are to be published (replace ‘xxx’ with actual accession numbers). It would also be good to provide accession numbers for the previously published transcriptome assembly and transcript count data.

To determine substrate specificity, it would have been better to determine enzyme kinetic properties, especially given that simple photometric assays are available to test for 4CL activity. This would have given a more accurate comparison compared to the competition assays used. However, I acknowledge that protein stability of purified 4CL enzymes is a frequently observed problem rendering it difficult to determine extensive kinetic properties.

You discuss that using a caffeate-specific 4CL would be an intuitive way to increase MbA production in Nicotiana and suggest targeted or directed evolution techniques to increase substrate-specificity. I agree but would suggest to start with a 4CL that already has a preference for caffeate; I think this is true for one of the Arabidopsis isoforms.

It remains unclear to me why it would be important to establish the early steps of flavonoid biosynthesis in yeast using the Montbretia enzymes when similar approaches using genes from other plants were already successful. Wouldn’t it be more efficient to build on these published successes and add the montbretin specific part to these strains that already produce basic flavonoids? I assume this relates to the ability to protect the engineered strains once or if efficient MbA production in yeast has been achieved making it commercially viable to produce it this way?

6. PLOS authors have the option to publish the peer review history of their article (what does this mean?). If published, this will include your full peer review and any attached files.

Reviewer #1: **Yes: **Bjoern Hamberger

Reviewer #2: No

---

## [Author Response · Author response to Decision Letter 0]

31 Aug 2021

please see the attached file: 00 Response to Reviewer comments 083021

---

## [Editor Report · Decision Letter 1]

2 Sep 2021

4-Coumaroyl-CoA Ligases in the Biosynthesis of the Anti-Diabetic Metabolite Montbretin A

PONE-D-21-23997R1

Dear Dr. Bohlmann,

We’re pleased to inform you that your manuscript has been judged scientifically suitable for publication and will be formally accepted for publication once it meets all outstanding technical requirements.

Kind regards,

Igor Cesarino, Ph.D

Academic Editor

PLOS ONE
---

## [Editor Report · Acceptance letter]

24 Sep 2021

PONE-D-21-23997R1 

4-Coumaroyl-CoA Ligases in the Biosynthesis of the Anti-Diabetic Metabolite Montbretin A 

Dear Dr. Bohlmann:

I'm pleased to inform you that your manuscript has been deemed suitable for publication in PLOS ONE. Congratulations! Your manuscript is now with our production department. 

Kind regards, 

on behalf of

Dr. Igor Cesarino 

Academic Editor

PLOS ONE